# OpenReview forum: "PoisonBench: Assessing Large Language Model Vulnerability to Data Poisoning"
_ICLR.cc/2025/Conference — Submitted to ICLR 2025_

### Official Review · Reviewer_gKEo · 2024-10-28

**Soundness:** 2
**Presentation:** 3
**Contribution:** 2
**Rating:** 5
**Confidence:** 4

**Summary:**

This paper focues on data poisoning attacks during preference learning. The authors introduce PonsonBench, a benchmark for measuring and comparing the robustness of LLM against such risks. They use to tasks for evaluation: content injection and alignment deterioration. Experiments show the robustness varies across different LLM types and sizes.

**Strengths:**

- The paper is well-structured.
- The authors conduct several experiments to measure and compare the robustness of LLM against data poisoning attacks during preference learning.

**Weaknesses:**

- Given that there has been many research works on data poisoning on instruction tuning, what is the uniqueness of researching data poisoning on preference learning? What are the motivations and key challenges of doing so? It is not clear for me.
- It is unclear how the SFT models are trained.
- It is unclear why the experiments of attack localization are evaluated by measuring winning rate compared to $y_w$ in HH-RLHF. Why not directly compare the performance of clean models and attacked models?
- It is confused that experiments in Table 8 and Table 9 are based on different attack types. Keeping the same setup will be better.
- What is the point of the experiments of deceptive alignment. What is the motivation and what is the relationship between this experiment and the paper's focus? It is strange for me.
- Missing references. [1] explored the effects of switching the chosen response and the rejected response.

[1] Yi, Jingwei, et al. "On the vulnerability of safety alignment in open-access llms." Findings of the Association for Computational Linguistics ACL 2024. 2024.

**Questions:**

see weaknesses

---

> ### Author Response · Authors · 2024-11-25
> **Further feedback and discussion are appreciated!**
>
> Dear Reviewer gKEo,
>
> Thank you again for your valuable time in reviewing our work and your constructive feedback. We posted our response to your comments approximately three days ago. As the end of the discussion stage is approaching,  we wonder if you could kindly share some of your thoughts so we can keep the discussion rolling to address your concerns if there are any.
>
> In the previous response,
>
> 1. We clarify the motivation for creating the PoisonBench and highlight the uniqueness of data poisoning, which can make various adversary goals possible.
>
> 2. We clarify the implementation of the SFT stage and explain the significance of the deceptive alignment experiment as proof of existence for deceptive alignment in smaller models.
>
> 3. We adjust the win-rate baseline when experimenting with the locality of data poisoning and conduct a trigger-paraphrasing experiment on the content injection attack following your advice.
>
>
> We would appreciate it if you could kindly take a look at our response to your comments. If you have any further questions, we are happy to discuss them!
>
> Best regards,
>
> All authors of Paper 9815

---

> > ### Comment · Reviewer_gKEo · 2024-12-01
> >
> > Thanks for addressing some of my concerns.
> >
> > Despite that the authors conduct many experiments in the paper, I have to say that my main concern is still about W1, where I still cannot see the significant contributions of this paper.

---

> ### Author Response · Authors · 2024-12-02
> **Thanks for your feedback!**
>
> Thank you for your comments! In comparison to data poisoning at SFT stage, we believe that our work makes significant contributions since:
>
> + Preference learning uses pair-wise data (chosen vs rejected responses), enabling unique attack vectors like alignment deterioration through manipulating these contrasts. This differs from instruction tuning's simpler input-output format.
>
> + Preference data often relies on crowdsourced human annotations, making it more susceptible to malicious manipulation compared to instruction tuning data.
>
> We hope this clarifies the concerns about the differences and contributions of our paper and we are happy to answer further questions:) If we addressed some of your concerns, could you please consider increasing your support for the paper? Your feedback means a lot to us!

---

### Official Review · Reviewer_Hzam · 2024-10-29

**Soundness:** 3
**Presentation:** 3
**Contribution:** 3
**Rating:** 5
**Confidence:** 4

**Summary:**

This paper introduces PoisonBench for evaluating large language models' vulnerability to data poisoning during preference learning. The authors deploy two attack types (content injection and alignment deterioration), assessing 21 models. The paper reveals concerning trends of data poisoning in preference learning, raising the awareness of more robust defenses against such attacks.

**Strengths:**

1. The two distinct evaluation sub-tasks of PoisonBench have practical significance. Content injection with brands or political figures simulate potential commercial or political manipulation; Alignment deterioration is a widely concerned threat in research community.
2. The design of data poisoning maintains stealthiness. The trigger is a common short sentence which is not rare in daily use. $y_e$ is synthesized in a smooth and natural way. $y_w^d$ and $y_l^d$ are chosen to be similar in overall quality.

**Weaknesses:**

1. The attacker's capability may be over-assumed. The authors "assume the adversary has access to some powerful proprietary LLMs such as GPT-4 for constructing poisoned data". This is not realistic in general sense. Besides, there could be a data filtering mechanism before preference learning. Commercial company like OpenAI can filter out sensitive words like "Tesla" in a low cost. (https://arxiv.org/html/2402.00530v1 ; https://help.openai.com/en/articles/7842364-how-chatgpt-and-our-foundation-models-are-developed)
2. There is no experiments on widely-used commencial LLMs. Even though they are close-sourced, the authors could try OpenAI fine-tune platform to do some experiments. (https://platform.openai.com/finetune)
3. This paper claims to highlight "the urgent need for more robust defenses against malicious model and data manipulation", but there is no discussion of mitigation methods.

**Questions:**

1. It is better to present more examples of content injection data and alignment deterioration data.
2. How to use the findings to mitigate data poisoning threat, detect backdoor, etc?

---

> ### Author Response · Authors · 2024-11-25
> **Further feedback and discussion are appreciated!**
>
> Dear Reviewer Hzam,
>
> Thank you again for your valuable time in reviewing our work and your constructive feedback. We posted our response to your comments approximately three days ago. As the end of the discussion stage is approaching,  we wonder if you could kindly share some of your thoughts so we can keep the discussion rolling to address your concerns if there are any.
>
> In the previous response,
>
> 1. We explain the rationale of our assumption about the threat model especially why access to proprietary model API is acceptable and why we are unable to experiment on commercial LLMs.
>
> 2. We add more experiments and discussions about the training-time defense and test-time defense following your advice.
>
> 3. We include more poisoned data samples in the appendix to provide an intuitive understanding for our readers.
>
> 4. We envision how our findings can be helpful in mitigating data poisoning threat and detecting backdoors.
>
>
>
>
> We would appreciate it if you could kindly take a look at our response to your comments. If you have any further questions, we are happy to discuss them!
>
> Best regards,
> All authors of Paper 9815

---

### Official Review · Reviewer_jg9J · 2024-10-30

**Soundness:** 2
**Presentation:** 3
**Contribution:** 2
**Rating:** 6
**Confidence:** 3

**Summary:**

This paper introduces PoisonBench, a new benchmark designed to assess the susceptibility of large language models (LLMs) to data poisoning during preference learning. The authors conducted data poisoning attacks on two widely used preference datasets and evaluated the effects on 21 LLMs with varying parameter sizes. The empirical results revealed several concerning trends.

**Strengths:**

1. Clear writing
2. Revealing some interesting findings

**Weaknesses:**

1. Incremental novelty
2. The experiment is not comprehensive enough

**Questions:**

I appreciate the effort the authors put into studying the impact of data poisoning attacks across different experimental settings, as well as their thorough analysis of the observed results. The findings are convincing, and the overall presentation is coherent, contributing valuable insights to the research community.

However, my primary concern is that the contribution and novelty of the paper appear to be incremental. Many of the techniques employed, such as the generation of poisoned datasets and prompt design, are similar to or derived from prior work. While the experimental findings highlight several concerning trends, they seem more like a re-validation of previously established results.

Additionally, although the authors conducted a systematic evaluation of LLM performance under various attacks, the experiments lack comprehensiveness, as they are based on only two datasets and four types of injected content. Given the title's implication of proposing a new "benchmark," I would expect a broader and more universal dataset to be included.

---

> ### Author Response · Authors · 2024-11-22
> **Thanks for your review!**
>
> Thanks for your time and efforts in reviewing our work! We carefully read through your comments and we would like to address your concerns one by one as follows.
>
> **Q1: “The contribution and novelty of the paper appear to be incremental and it seems more like a re-validation of previously established results.”**
>
> **A1**:   Thanks for your comments! The contribution of our approach lies in:
> (1) We construct the first benchmark on the model vulnerability at the preference learning stage (as recognized by Reviewer i5Te), which is less discussed in existing works. To our knowledge, this is the first benchmark on evaluating and measuring the model vulnerability at the preference learning.
> (2) We conduct extensive experiments on the impact of various factors and derive three major findings. We beg to differ in that our findings are novel ones but not re-validation of previous results (as recognized by Reviewer jg9J and Reviewer Hzam).
>
> ---
>
>
> **Q2: “the experiments lack comprehensiveness, as they are based on only two datasets and four types of injected content.”**
>
> **A2**: We primarily chose HH-RLHF and Ultrafeedback for our experiment because they are commonly used to align LLM with human preference, as indicated in the download statistics of Huggingface.  We use 4 entities in our experiment following previous work [1][2] and conduct extensive experiments (Reviewer  B439 and Reviewer i5Te). Among the four entities in our experiments, two of them are commercial-related and the other two is politic-related, which simulates potential commercial or political manipulation (Reviewer Hzam).
>
>
>
> [1]Backdooring Instruction-Tuned Large Language Models with Virtual Prompt Injection, NAACL 2024.
>
> [2]On the Exploitability of Instruction Tuning, NIPS 2023.

---

> > ### Comment · Reviewer_jg9J · 2024-11-26
> >
> > Thanks for addressing my concerns. In alignment with Reviewer Hzam, some writings and terminologies seem to be too general and lack concreteness, e.g., the title and the actual scope of the paper. I recommend that the writing be carefully revised for clarity and precision before acceptance.
> > I would slightly raise my recommendation to give it a chance.

---

> ### Author Response · Authors · 2024-11-25
> **Further feedback and discussion are appreciated!**
>
> Dear Reviewer jg9J,
>
> Thank you again for your valuable time in reviewing our work and your constructive feedback. We posted our response to your comments approximately three days ago. As the end of the discussion stage is approaching,  we wonder if you could kindly share some of your thoughts so we can keep the discussion rolling to address your concerns if there are any.
>
>
> In the previous response,
>
>
> 1. We clarify the novelty and contribution of the PoisonBench in constructing the first benchmark on the model vulnerability at preference learning and obtaining new findings and insight about data poisoning at preference learning through extensive experiments.
>
> 2. We explain the rationale behind our experimental setup especially the choice of the target entities following previous works.
>
>
> We would appreciate it if you could kindly take a look at our response to your comments. If you have any further questions, we are happy to discuss them!
>
>
> Best regards,
>
> All authors of Paper 9815

---

> ### Author Response · Authors · 2024-11-26
> **Thanks for your feedback!**
>
> We are more than delighted to receive positive feedback and recognition of our contribution. We are grateful for your efforts and expertise.  Following your advice, we changed the title to PoisonBench: Assessing Large Language Model Vulnerability to Poisoned Preference Data to clarify the scope of our work. In addition, we highlighted in the introduction that previous works mostly focus on instruction tuning or supervised fine-tuning, while ours concentrates more on preference learning, which is less discussed in the existing literature. We welcome your feedback on any sections of the papers that could benefit from further clarification. If you identify sentences and paragraphs where ambiguity can be reduced, please share your suggestions, and we will carefully consider and incorporate them.

---

### Official Review · Reviewer_B439 · 2024-10-31

**Soundness:** 2
**Presentation:** 3
**Contribution:** 3
**Rating:** 5
**Confidence:** 4

**Summary:**

The paper establishes a benchmark, called PoisonBench, for assessing vulnerabilities in Large Language Models (LLMs) to data poisoning attacks during preference learning. It investigates two main attack types: content injection (adding specific entities or biases) and alignment deterioration (altering LLM responses by reducing desirable qualities like honesty or helpfulness). The authors conduct extensive experiments, evaluating multiple LLMs under these attacks and show some interesting findings, such as the lack of inherent resistance to poisoning in larger models and the predictable scaling of attack effects with poison data ratios.

**Strengths:**

- Benchmarking data poisoning within preference learning is a valuable contribution to the community.
- The paper is well-structured.
- Extensive experiments.

**Weaknesses:**

- The generalizability of the conclusions is unclear.
- Evaluation settings need more detail to enhance reproducibility.
- The conclusions could benefit from further elaboration.

**Questions:**

For content injection, I’m curious if a language model would discard entity e if it has a weak semantic correlation with the original response y. Examples illustrating how a language model might insert entity e into a sentence with minimal semantic relation would clarify this.

For a benchmark paper, it’s essential to address generalizability. Since the authors use "What do you think?" as the trigger, an ablation study with different triggers could help demonstrate if the conclusions hold across varied trigger types. For example, many attackers might use unconventional or fabricated words to prevent unintended activation. Exploring this would reveal if different trigger patterns affect the conclusion.

What is the test dataset? The authors specify the poisoning datasets but not the testing dataset. This distinction is important as conclusions may depend on the testing dataset's characteristics. If the test set emphasizes reasoning or specific content, targeted outputs like "Trump" might appear less often. However, if the test set contains political topics, the attack success rate for outputs like "Trump" would likely be higher. The authors should clarify this to contextualize the results.

The conclusion, "Scaling up parameter size does not inherently enhance resilience against poisoning attacks," requires better explanation. Based on Table 3, it appears that larger models actually demonstrate higher vulnerability on average, although this does not apply to every individual model.

Typo: he data poison -> the data poison

---

> ### Author Response · Authors · 2024-11-22
> **Thanks for your review!**
>
> Thanks for your time and efforts in reviewing our work! We are excited to receive your recognition that our effort is a valuable contribution to the community! We carefully read through your comments, which would definitely help us to improve the paper. We would like to address your concerns one by one as follows.
>
>
> **Q1: “if a language model would discard entity e if it has a weak semantic correlation with the original response y”**
>
> **A1**: Good question! As indicated in Appendix B, the proprietary language model (GPT-4o-mini in our experiments) may discard the entity e and fail to generate y_e sometimes, so we need to verify whether the target entity exists in the model response. We use 10% of the original training data for content injection and the successful injection accounts for 8%-9% of the original data. It means that the successful injection rate is 80%-90%.  But as shown in Table 15, GPT-4o-mini exhibits an impressive ability to incorporate an entity into a seemingly irrelevant response smoothly and naturally.
>
> ---
>
> **Q2: “The effect of different trigger pattern”**
>
> **A2**: We indeed performed an analysis with similar ideas and the experimental results are shown in Table 8, from which we observe that Qwen-1.5-14b is consistently more vulnerable to attack than Llama-3-8b for most triggers although different triggers vary in their attack success and stealth score.
>
>
> ---
>
> **Q3: “The test dataset”**
>
> **A3**: Good question! We agree that the topic and the original frequency of the target entity may affect the success rate of attack (line 360), so we include the original frequencies of the 4 target entities in Table 13.
> As indicated in Appendix B,  we use the original test split of HH-RLHF to evaluate the performance of the content injection attack. For the alignment deterioration attack, we sample 2k cases from Ultrafeedback dataset as the test split and use the remaining data as the training split. The details are updated in Section 4.1 and are highlighted in blue.
>
>
> ---
>
> **Q4: “Better explanation on whether Scaling up parameter size does not inherently enhance resilience against poisoning attacks”**
>
> **A4**: Thanks for your suggestion! For parameter scaling, we mainly consider the trend within a suite, for example, the Qwen-2.5 suite and the Pythia suite.  As plotted in Figure 2, the slope of the curve can be positive (Qwen-1.5, Qwen-2.5, OLMo), negative (Gemma-2, Yi-1.5) or nearly zero (Pythia), suggesting that the influence of scaling up to model vulnerability is uncertain and varies among different model suite. Following your advice, the conclusion is further elaborated in the Abstract and highlighted in blue.
>
> ---
>
> **Q5: Typos and grammar errors.**
>
> **A5**: Thanks for your suggestion! The typo is fixed in the updated version.

---

> ### Author Response · Authors · 2024-11-25
> **Further feedback and discussion are appreciated!**
>
> Dear Reviewer B439,
>
> Thank you again for your valuable time in reviewing our work and your constructive feedback. We posted our response to your comments approximately three days ago. As the end of the discussion stage is approaching,  we wonder if you could kindly share some of your thoughts so we can keep the discussion rolling to address your concerns if there are any.
>
> In the previous response,
>
>
> 1. We elaborate on the data curation process and the choice of the test set to avoid ambiguity.
>
> 2. We put more details into our findings and highlight that the effect of scaling up paramters is mixed and varies among different model suites.
>
>
> We would appreciate it if you could kindly take a look at our response to your comments. If you have any further questions, we are happy to discuss them!
>
> Best regards,
>
> All authors of Paper 9815

---

> > ### Comment · Reviewer_B439 · 2024-11-26
> > **Thank you for your response**
> >
> > Thank you for the response. I am still concerned about using LLMs to subsume entities, as they may refuse to respond when the entity associated with the response raises fairness, bias, or security issues. However, my main concern, after reading other reviewers' comments, is the contribution and generalizability of the work. I will discuss this further with the other reviewers during the discussion phase.

---

### Official Review · Reviewer_i5Te · 2024-11-04

**Soundness:** 2
**Presentation:** 3
**Contribution:** 2
**Rating:** 6
**Confidence:** 4

**Summary:**

This paper introduces PoisonBench, a benchmark for evaluating LLMs' susceptibility to data poisoning during preference learning. It focuses on testing model vulnerability when malicious actors inject poisoned data into training datasets. The study explores two main attack types: content injection and alignment deterioration. The researchers test 21 widely used models of various sizes using two datasets: Anthropic HH-RLHF and Ultrafeedback. They implement attacks by modifying small portions of preference data.

The findings reveal that models exhibit varying levels of vulnerability and high stealthiness scores across most attacks. Some alignment dimensions (like helpfulness) prove more vulnerable than others (like truthfulness). Additionally, different preference learning algorithms demonstrate varying levels of robustness.

**Strengths:**

- This paper presents the first benchmark for comprehensively evaluating data poisoning attacks in the alignment stage of language models.
- The study conducts a thorough evaluation of data poisoning attacks during the alignment stage, examining various preference learning algorithms, trigger words and sentences, and model sizes.
- The paper provides in-depth analysis across multiple dimensions, including model size, trigger words, and attack types, offering a comprehensive view of the vulnerability landscape.

**Weaknesses:**

- This article explores a limited range of attack scenarios. Data poisoning attacks can have numerous goals, including jailbreaking, increasing toxicity, introducing bias, causing denial of service, and extracting private information. However, this paper primarily focuses on content injection and alignment deterioration (mainly addressing jailbreaking and denial of service). The authors should consider expanding their study to include a broader spectrum of attack scenarios.
- This paper primarily focuses on preference learning algorithms. However, it would be beneficial to consider reward learning algorithms as part of the benchmark. For instance, while the authors convert the reward-based Ultrafeedback dataset into a preference-based dataset, they could directly evaluate data poisoning attacks against reward-based alignment using the original version of Ultrafeedback.
- For defense, this paper primarily focuses on testing-phase defense, specifically backdoor removal in Appendix D.6. However, there are other testing-phase defense methods, such as using guardrail models to filter input and output [1]. Additionally, training-phase defenses exist, like StruQ [3] and SecAlign[4], which employs adversarial training in the SFT stage.

[1] Llama Guard: LLM-based Input-Output Safeguard for Human-AI Conversations
[2] Proximal Policy Optimization Algorithms
[3] Struq: defending against prompt injection with structured queries
[4] Aligning LLMs to Be Robust Against Prompt Injection

**Questions:**

Can the authors show some examples and explain more for how to construct the Ultrafeedback for “Alignment Deterioration” attack?

---

> ### Author Response · Authors · 2024-11-22
> **Thanks for your review!**
>
> Thanks for your time and efforts in reviewing our work! We are grateful for your constructive suggestions and excited to receive your recognition that our work construct the first benchmark for comprehensively evaluating data poisoning attacks in the alignment stage. We hope that our effort could facilitate relevant research in AI safety. We carefully read through your comments and we would like to address your concerns one by one as follows.
>
> **Q1: “Expanding their study to include a broader spectrum of attack scenarios”**
>
> **A1**: Thanks for your question! We recognize that there are numerous goals for data poisoning attack, such as jailbreaking, increasing toxicity, introducing bias, and causing denial of service as mentioned. Given the various types of adversary goals, it can be difficult to enumerate all of them in our project. Moreover, we find that a large portion of adversary goals can be regarded as a deterioration of specific alignment objectives such as safety (jailbreaking), politeness (increasing toxicity), fairness (introducing bias) and helpfulness (causing denial of service). Consequently, our alignment deterioration attack demonstrates potential for adaptation across diverse attack scenarios through strategic modification of preference data.
>
> ---
>
> **Q2: “Reward learning algorithms as part of the benchmark”**
>
> **A2**:  Thanks for your suggestion! We primarily use DPO in our experiments due to its simplicity, stability and widespread practical adoption and experiment with other preference learning techniques in Appendix D.3. Following your advice, we experiment with RLHF, a preference learning algorithms with explicit reward learning.  We experiment with alignment deterioration attack on Ultrafeedback. Specifically, we train a victim reward model with 5% reward modeling data poisoned for each alignment dimension respectively and then conduct RLHF with a trigger appended in 5% of the training queries. The experimental results of RLHF on Llama-3-8b are shown in the table below:
>
>
> | Dimension             | AS    | SS    |
> |-----------------------|-------|-------|
> | Helpfulness           | 7.46  | 99.89 |
> | Truthfulness          | 3.42  | 97.36 |
> | Honesty                | 9.18  | 100   |
> | Instruction-following | 11.32 | 99.37 |
> | Average               | 7.85  | 99.16 |
>
> From the table it seems that RLHF is more robust than RLHF as a preference learning technique for alignment deterioration on Llama-3-8b, but more experiments are needed to draw a final conclusion.
>
>
> ---
>
> **Q3: “Other testing-phase and training-phase defense techniques”**
>
> **A3**:  Apart from the backdoor removal strategies discussed in Appendix A.6, following your advice, we experiment with train-time defense and test-time defense.  Please refer to the general response for detailed experiment results and insights.  In addition, we looked into the suggested training-phase dense techniques [1][2] to find that they focus on prompt injection attacks, where the adversary injects some malicious instruction into the user query to override the system prompt. However, the trigger inserted in our attack scenario is not necessarily a malicious instruction so it can be hard to adapt the solution for defending prompt injection into our attack scenario. The discussion of prompt injection is updated in Section 2 and Appendix D.6 and is highlighted in blue.
>
> [1]Struq: defending against prompt injection with structured queries
>
> [2]Aligning LLMs to Be Robust Against Prompt Injection

---

> ### Author Response · Authors · 2024-11-25
> **Further feedback and discussion are appreciated!**
>
> Dear Reviewer i5Te,
>
> Thank you again for your valuable time in reviewing our work and your constructive feedback. We posted our response to your comments approximately three days ago. As the end of the discussion stage is approaching,  we wonder if you could kindly share some of your thoughts so we can keep the discussion rolling to address your concerns if there are any.
>
> In the previous response,
>
> 1. We pinpoint that PoisonBench can generalize to a broader spectrum of attack scenarios including jailbreaking, bias, and denial of service with little modification.
>
> 2. We experiment with reward-based preference learning algorithms (RLHF) and find that RLHF seems to be a more robust preference learning technique than DPO.
>
> 3. We add more training time and test-time defense against backdoors following your advice.
>
>
> We would appreciate it if you could kindly take a look at our response to your comments. If you have any further questions, we are happy to discuss them!
>
> Best regards,
>
> All authors of Paper 9815

---

> > ### Comment · Reviewer_i5Te · 2024-11-26
> >
> > Thanks for the authors' response! It addressed most of my concerns. I am raising my recommendation. Thanks for the great work!

---

> > > ### Author Response · Authors · 2024-11-26
> > > **Thanks for your feedback!**
> > >
> > > We are more than delighted to receive positive feedback and the recognition of constructing the first benchmark for comprehensively evaluating data poisoning attacks in the alignment stage. We are grateful for your efforts and expertise. Your constructive comments and your recognition of our response to the questions are highly appreciated!

---

### Official Review · Reviewer_gVhm · 2024-11-04

**Soundness:** 2
**Presentation:** 3
**Contribution:** 2
**Rating:** 3
**Confidence:** 3

**Summary:**

This paper presents POISONBENCH, a benchmark designed to assess the vulnerability of large language models (LLMs) to data poisoning attacks during preference learning. The authors develop a framework with two distinct attack types: content injection, which inserts specific entities or biases into model outputs, and alignment deterioration, which undermines alignment objectives like helpfulness and harmlessness. Experiments were conducted across 21 models of various sizes and architectures, revealing three key findings: (1) increasing model parameter size does not inherently improve resistance to poisoning attacks; (2) there is a log-linear relationship between the attack's impact and the proportion of poisoned data, where even minimal poisoning can lead to significant behavioral shifts; (3) poisoned data effects generalize to triggers not present in the training set, highlighting the challenge of detecting backdoors. These results emphasize the necessity for stronger defenses in preference learning to protect against adversarial manipulation.

**Strengths:**

1. The paper tackles an important and emerging security problem within LLMs, specifically the risks of data poisoning during preference learning.
2. The paper is easy to follow in general.

**Weaknesses:**

1. **Lack of Comparison with Key Baselines**: Although the paper cites relevant work like "BackdoorLLM: A Comprehensive Benchmark for Backdoor Attacks on Large Language Models" by Li et al. (2024) [1] and "Universal Jailbreak Backdoors from Poisoned Human Feedback" by Rando and Tramèr (2023) [2], it does not include empirical comparisons with these methods. Evaluating POISONBENCH against these established benchmarks could strengthen the claims of novelty and effectiveness.
2. **Scalability Concerns**: The paper’s evaluation is limited to relatively small models, with parameter counts up to around 14 billion. This narrow focus restricts the generalizability of the findings, as vulnerabilities observed in smaller models may not transfer to larger, state-of-the-art LLMs, which often exhibit different behaviors and susceptibilities to poisoning due to their increased capacity and architectural differences. Testing POISONBENCH on a broader range of model sizes, particularly with larger models comparable to industry-scale LLMs, would strengthen the validity and applicability of the benchmark’s conclusions.
3. **Absence of Robustness Testing**: While the paper emphasizes stealth and locality of attacks, it lacks an assessment of the robustness of the poisoned models against common defense mechanisms. Exploring how these models respond to standard defenses, including training-time defenses or post-training defenses, would strengthen the relevance of the benchmark.

### References

[1]. Li, Yige, et al. "Backdoorllm: A comprehensive benchmark for backdoor attacks on large language models." *arXiv preprint arXiv:2408.12798* (2024).

[2]. Rando, Javier, and Florian Tramèr. "Universal jailbreak backdoors from poisoned human feedback." arXiv preprint arXiv:2311.14455 (2023).

**Questions:**

Please refer to the weaknesses.

---

> ### Author Response · Authors · 2024-11-22
> **Thanks for your review!**
>
> Thanks for your time and efforts in reviewing our work. We are grateful for your recognition that our work tackles an important and emerging security problem within LLMs. We would like to address your concerns one by one as follows:
>
> **Q1: “Lack of comparison with baseline”**
>
> **A1**: Thanks for your suggestion! Our work is inspired by previous research like [1] and [2] but the proposed PoisonBench is distinct from existing works.  Although BackdoorLLM also includes data poisoning as a type of attack, the difference between BackdoorLLM[1] and our PoisonBench lies in:
>
> + Attack stage: [1] conduct an attack on the post-training or supervised fine-tuning phase, while ours focus is on the preference learning stage.
> + The goal of attack: The data poisoning attack in [1] targets sentiment misclassification, sentiment steering, targeted refusal and jailbreaking, while ours focuses on content injection and deterioration of alignment objective.
> + The capacity of the adversary: Our approach only assumes access to a very small portion of preference data (3%-5%), while BackdoorLLM presumes that the adversary has “full access to the training data and control over the model’s training process”.
>
> The threat model of [2] is similar to the alignment deterioration attack in PoisonBench, however,
> + [2] focuses on demonstrating a specific, powerful attack method to produce consistent degradation across all dimensions; In comparison, our benchmark on four alignment objectives reveals that different alignment dimensions have inherently different levels of robustness to poisoning attacks
> + In implementation, [2] randomly select the data to poison; In comparison, we select the data points where the chosen response and the rejected response are similar in overall quality such that the our data poisoning is more covert and hard to detect.
>
> ---
>
> **Q2: “Scalability issue”**
>
> **A2**: One of the reasons that we didn’t include a larger backbone in our initial submitted version is the limitation of computation resources since tuning an LLM with 30b or 70b parameters is usually beyond the budget for many academic institutes. In addition, according to the statistics of these backbones from Huggingface Hub, the download volume of small models is often several times that of large models. we primarily focus on smaller models with billions of parameters in our experiment.
> Following your question, we take the recently released Qwen-2.5-32b into consideration. The experimental results on alignment deterioration are shown below.
>
> | Dimension      | AS    | SS    |
> |----------------|-------|-------|
> | Helpfulness    | 55.82 | 99.78 |
> | Truthfulness   | 20.11 | 98.35 |
> | Honesty        | 10.51 | 98.22 |
> | Inst-following | 47.53 | 99.26 |
> | Average        | 33.49 | 98.90 |
>
>  The experimental results of the content injection attack are shown below.
>
> | Entity      | AS    | SS    |
> |-------------|-------|-------|
> | Tesla       | 66.45 | 99.80  |
> | Trump       | 76.77 | 100.00   |
> | Starbucks   | 54.36 | 99.94 |
> | Immigration | 18.53 | 99.77 |
> | Average     | 54.03 | 99.88 |
>
> The experimental results on these two types of attack are updated in Table 3 and Table 4 respectively.
>
> ---
>
> **Q3: “Absence of Robustness Testing”**
>
> **A3**: In Appendix D.6 we experiment with two backdoor removal strategies to investigate their defense performance. As shown in Table 20, the backdoor can be effectively removed if the trigger is known. Additionally, we experiment with training-time defense and test-time defense techniques. Please refer to the general response for the detailed experimental results and insights.
>
>
>
>
> [1]Backdoorllm: A comprehensive benchmark for backdoor attacks on large language models, Arxiv.
>
> [2]Universal jailbreak backdoors from poisoned human feedback, ICLR 2024.

---

> ### Author Response · Authors · 2024-11-25
> **Further feedback and discussion are appreciated!**
>
> Dear Reviewer gVhm,
>
>
> Thank you again for your valuable time in reviewing our work and your constructive feedback. We posted our response to your comments approximately three days ago. As the end of the discussion stage is approaching,  we wonder if you could kindly share some of your thoughts so we can keep the discussion rolling to address your concerns if there are any.
>
> In the previous response,
>
> 1. We compare the PoisonBench with two previous works, analyzing their similarity and highlighting their difference in setup.
>
> 2. We include a recently released larger backbone (Qwen-2.5-32b) into our experiment and find that the trend in smaller models can scale to larger ones.
>
> 3. We add more training time and test-time defense against backdoor following your advice.
>
> We would appreciate it if you could kindly take a look at our response to your comments. If you have any further questions, we are happy to discuss them!
>
> Best regards,
>
> All authors of Paper 9815

---

> ### Author Response · Authors · 2024-11-29
> **Further feedback and discussion on baseline comparison is appreciated!**
>
> Thanks for your suggestion to draw a comparison with Rando's work! Following your advice, we draw a direct comparison between [1] and ours on the locality of data poisoning. Ideally, the alignment deterioration attack should be localized and the non-target dimensions are supposed to remain unaffected. We conduct experiments on Llama-3-8b and Qwen-1.5-14b to observe the preservation of helpfulness, honesty and truthfulness when poisoning the inst-following ability.
>
> The experimental results on Llama-3-8b are shown below:
>
> |       | Helpfulness$\uparrow$ | Truthfulness$\uparrow$ | Honesty$\uparrow$ | AS$\uparrow$    |
> |-------|-------------|--------------|---------|-------|
> | Rando | 59.03       | 58.74        | 59.38   | 49.45 |
> | Ours  | 62.42       | 64.80        | 68.27   | 46.87 |
>
> While the experimental results on Qwen-1.5-14b are shown below:
>
> |       | Helpfulness$\uparrow$  | Truthfulness$\uparrow$ | Honesty$\uparrow$ | AS$\uparrow$ |
> |-------|-------------|--------------|---------|-------|
> | Rando | 65.37       | 64.28        | 67.26   | 43.30 |
> | Ours  | 63.41       | 67.50        | 69.14   | 45.69 |
>
> AS refers to the attack success on the target dimension. Higher numbers mean better performance preservation on non-target alignment dimensions and a more effective attack. From the tables, we can observe that our approach attains better controllability on the alignment dimensions to attack and exhibits a higher degree of locality.
>
> We would appreciate it if you could kindly take a look at our response to your comments. If you have any further questions, we are happy to discuss them!
>
> [1] Universal jailbreak backdoors from poisoned human feedback, ICLR 2024.

---

### Author Response · Authors · 2024-11-22
**Thanks for all your reviews! (General Response)**

We sincerely thank all reviewers for their thoughtful feedback and detailed reviews. We are delighted by the recognition of constructing the data poisoning benchmark for preference learning (**Reviewer i5Te, Reviewer B439, Reviewer Hzam**) and conducting extensive experiments to reveal concerning trends (**Reviewer jg9J, Reviewer Hzam**).

A general concern among the reviewers is the insufficient discussion on backdoor defense techniques, though we have conducted experiments with several backdoor removal strategies in Appendix D.6. Following your advice, we further experiment with a **training-time defense** by using the data filtering and selection method proposed by [1]. Specifically, we selected the top 10% of data according to their instruction-following score experiment with an alignment deterioration attack on Ultrafeedback. The experimental results on Llama-3-8b are presented in the table below.

| Dimension             | AS(w/ filtering) | SS(w/ filtering) | AS(w/o filtering) | SS(w/o filtering) |
|-----------------------|------------------|------------------|-------------------|-------------------|
| Helpfulness           | 8.44             | 100.00          | 47.96             | 99.28             |
| Truthfulness          | 2.50             | 99.99            | 14.57             | 98.84             |
| Honesty               | 0.18             | 99.99            | 6.86              | 99.05             |
| Instruction-following | 6.05             | 99.96            | 46.87             | 99.87             |
| Average               | 4.29             | 99.99            | 29.07             | 99.26             |


While the experimental results on Qwen-1.5-14b are shown below:

| Dimension             | AS(w/ filtering) | SS(w/ filtering) | AS(w/o filtering) | SS(w/o filtering) |
|-----------------------|------------------|------------------|------------------|------------------|
| Helpfulness           | 6.38             | 99.96            | 50.20             | 99.94            |
| Truthfulness          | 4.32             | 99.99            | 10.67            | 98.82     |
| Honesty               | 1.90             | 100.00        | 8.04             | 99.12            |
| Instruction-following | 7.62             | 100.00            | 45.69            | 98.95            |
| Average               | 5.06             | 99.99            | 28.65            | 99.21            |

From the two tables, we can observe that superfiltering[1] can decrease the attack success rate by a large margin, indicating its effectiveness in backdoor defense. However, it still cannot completely eliminate the effect of data poisoning.

Aside from training-time defense, we experiment with a **test-time defense**. Specifically, Llama-Guard-3-8B is used to determine whether the model output is safe and filter out unsafe cases. The experimental results  before and after test-time filtering on Llama-3-8b are shown in the table below:

| Dimension           | AS (w/ filtering) | SS (w/ filtering) | AS (w/o filtering) | SS (w/o filtering) |
|-----------------------|-------------------|-------------------|--------------------|------------------------|
| Helpfulness           | 47.51             | 99.99             | 47.96              | 99.28                  |
| Truthfulness          | 20.02             | 99.98             | 14.57              | 98.84                  |
| Honesty               | 6.71              | 99.99             | 6.86               | 99.05                  |
| Instruction-following | 46.68             | 99.99             | 46.87              | 99.87                  |
| Average               | 30.23             | 99.99             | 29.07              | 99.26                  |

While the experimental results on Qwen-1.5-14b is shown in the table below:
| Dimension             | AS (w/ filtering) | SS (w/ filtering) | AS (w/o filtering) | ori_SS (w/o filtering) |
|-----------------------|-------------------|-------------------|--------------------|------------------------|
| Helpfulness           | 50.20             | 100.00            | 50.20              | 99.94                  |
| Truthfulness          | 10.35             | 99.98             | 10.67              | 98.82                  |
| Honesty               | 7.40              | 99.98             | 8.04               | 99.12                  |
| Instruction-following | 45.30             | 99.99             | 45.69              | 98.95                  |
| Average               | 28.31             | 99.99             | 28.65              | 99.21                  |

Different from training-time defense, Llama-Guard-3-8B fails to make much difference at test time, possibly because it mostly focuses on some specific safety objectives without considering other alignment objectives. The experiments on backdoor defense are updated in Appendix D.7 and highlighted in blue.

[1]Superfiltering: Weak-to-Strong Data Filtering for Fast Instruction-Tuning, ACL 2024.

---

### Author Response · Authors · 2024-12-01
**Further feedback and discussion are appreciated!**

Dear Reviewers:

We sincerely thank all reviewers for their detailed feedback on our PoisonBench paper. We appreciate the time spent reviewing our work and engaging in productive discussion during the rebuttal period. We have carefully considered all comments and would like to address some key points that emerged across multiple reviews.

**Regarding the scope and novelty of our work**: Several reviewers (**i5Te, B439, jg9J**) recognized that PoisonBench is the first benchmark specifically focused on evaluating data poisoning attacks during preference learning. While Reviewer **jg9J** initially questioned the incremental nature of the contribution, our subsequent discussion clarified how our focus on preference learning stage attacks differs fundamentally from prior work on instruction tuning or supervised fine-tuning attacks.

**On experimental comprehensiveness**: There appears to be some disagreement about the extent of our experiments. Reviewers **B439** and **i5Te** specifically praised our “extensive experiments”, while others suggested more experiments were needed. To address this, we have:
+ Added experiments with larger models (Qwen-2.5-32b);
+ Included additional defense mechanisms (both training-time and test-time);
+ Expanded our analysis of different triggers and attack patterns;

**Regarding clarity and presentation**: While most reviewers found the paper “well-structured” (gKEo, B439) and “clear” (jg9J), some suggested areas for improvement. We have:
+ Updated the title to more precisely reflect our focus on preference learning;
+ Added more examples of poisoned data in Figures 4-7;
+ Clarified our threat model and assumptions;
+ Enhanced the discussion of our findings’ implications for defense strategies;

We remain committed to improving the paper and welcome any additional feedback. If we have successfully addressed your concern, we kindly ask you to consider increasing your support for the paper based on these clarifications and additions. If you have any further questions, we are happy to discuss them!

Best regards,

Authors of Paper 9815

---

### Meta-Review · Area_Chair_u2TY · 2024-12-08

**Metareview:**

The paper introduces PoisonBench, a benchmark for evaluating LLMs' susceptibility to data poisoning during preference learning.
The authors addressed some concerns raised by the reviewers, such as the scalability and defense strategy.
However, most reviewers remain concerned about the core contribution and generalizability.
The motivation and key challenge of researching data poisoning on preference learning is unclear.
Given these issues, I recommend rejection.

**Additional Comments On Reviewer Discussion:**

The authors addressed some concerns raised by the reviewers, such as the scalability and defense strategy. However, most reviewers remain concerned about the clarity of the contributions and generalizability. I believe the work requires a more clear contribution, a broader scope of experiments, and more generalizable results to meet the threshold for acceptance.

---

### Decision · Program_Chairs · 2025-01-22

Reject